# Characterisation and In Vitro Evaluation of Fenugreek (*Trigonella foenum-graecum*) Seed Gum as a Potential Prebiotic in Growing Rabbit Nutrition

**DOI:** 10.3390/ani10061041

**Published:** 2020-06-17

**Authors:** Jihed Zemzmi, Luis Ródenas, Enrique Blas, Taha Najar, Juan José Pascual

**Affiliations:** 1Materials, Molecules and Applications Laboratory, Preparatory Institute for Scientific and Technical Studies, University of Carthage, BP51, La Marsa 2070, Tunisia; zemzmijihed@gmail.com (J.Z.); najar.taha@inat.agrinet.tn (T.N.); 2Institute for Animal Science and Technology, Universitat Politècnica de València, Camino de Vera, s/n, 46022 Valencia, Spain; luiromar@dca.upv.es (L.R.); eblas@dca.upv.es (E.B.); 3Department of Animal Production, National Agronomic Institute of Tunisia, University of Carthage, 43 Avenue Charles Nicolle, Mahrajène 1082, Tunisia

**Keywords:** fenugreek seed, galactomannan, rabbit, digestion in vitro, prebiotic

## Abstract

**Simple Summary:**

A fenugreek seed gum, extracted from *Trigonella foenum-graecum* seeds and rich in galactomannan, was chemically and physically characterised and its prebiotic potential for young rabbits was evaluated in vitro, both as pure fenugreek seed gum and when included up to 20 g/kg in rabbit diets rich in soluble and insoluble fibre. Fenugreek seed gum was resistant to pepsin and pancreatin digestion but was totally fermented by rabbit caecal bacteria. Fenugreek seed gum linear inclusion up to 20 g/kg in diets rich in soluble fibre has led to a reduction in the solubility of some nutrients during in vitro enzymatic phase and an increase in the fermented fraction. Fenugreek seed gum satisfies two essential conditions of a prebiotic: resistance to enzymatic digestion and being totally fermented by caecal bacteria.

**Abstract:**

Some components of soluble fibre appear to have prebiotic effects that can contribute to improving digestive health in post-weaning rabbits. In this work, a fenugreek seed gum (FGS), extracted from *Trigonella foenum-graecum* seeds and rich in galactomannan, was characterised. Both the pure FSG and ten substrates obtained by the inclusion of 0, 5, 10, 15 and 20 g/kg of FSG in diets rich in soluble (SF) and insoluble (IF) fibre were evaluated in vitro to determine FSG prebiotic potential for rabbit diets. FSG was rich in total sugars (630 g/kg dry matter), consisting entirely of galactose and mannose in a 1:1 ratio, and a moderate protein content (223 g/kg dry matter). Pure FSG was affected very little by in vitro digestion, as only 145 g/kg of the FSG was dissolved during the enzymatic phase. However, the linear inclusion of FSG up to 20 g/kg in growing rabbit feeds has led to a reduction in the solubility of some nutrients during in vitro enzymatic phase, especially in SF diets. Pure FSG not digested during the enzymatic phase almost completely disappeared during the in vitro fermentation phase, 984 g/kg of this indigestible fraction. However, although linear inclusion of FSG up to 20 g/kg in SF diets increased the fermented fraction, no relevant changes in the fermentation profile were observed. In conclusion, FSG satisfies two essential conditions of the prebiotic effect, showing resistance to in vitro enzymatic digestion and being totally fermented in vitro by caecal bacteria, although in vivo studies will be necessary to determine its prebiotic potential.

## 1. Introduction

Digestive disorders are the main cause of post-weaning mortality and veterinary visits in rabbit farms, and epizootic rabbit enteropathy (ERE) remains the major cause of these losses [1,2]. The solution to these digestive disorders must come through control of causative microorganisms. However, we know that diet seems to be a risk factor, mainly due to its ability to modulate the microbiota [3]. It is well known that a deficit in fibre or an excess of protein increase the risk of digestive disorders [4,5], and that the level and nature of the fibre can also affect the digestive health of weaned rabbits, thus a minimum level of both insoluble and soluble fractions of fibre are required to reduce the digestive risk [3]. However, even when designing diets under the recommended standards, the need to use antimicrobials after weaning as a consequence of these digestive disorders is very frequent. Some of the substances that constitute the soluble fibre affect the digestive utilisation of soluble and insoluble fibre fractions, and their dietary inclusion is usually associated with reduced mortality in growing rabbits affected by epizootic rabbit enteropathy [6]. Furthermore, Falcão-e-Cunha et al. [7] reported that some prebiotics, which are normally part of the soluble fibre of some plants, could be alternatives to antibiotics in young rabbit nutrition. Some fructooligosaccharides, galactooligosaccharides and mannooligosaccharides have been tested on rabbits with inconsistent results as regards mortality, growth performance and feed conversion ratio [8,9,10,11,12]. However, these prebiotics seem to have the capacity to improve the barrier effect against pathogens, promoting recovery of the intestinal mucosa, increasing the production of total volatile fatty acids (VFA) and reducing the ammonia content in the rabbit caecum [13,14,15].

Among these members of the soluble fibre group there is a non-starch polysaccharide known as galactomannan, composed of a (1-4)-linked beta d-mannopyranose backbone with branch points from their 6-positions linked to alpha-d-galactose [16]. Galactomannan has been reported as a potential prebiotic for starting broilers when used with the appropriate exogenous enzymes [17], and seems to contribute to increased serum antibody against Newcastle disease virus in this species [18]. In pigs, 10% of dietary carob tree seeds as a source of locust bean gum, rich in galactomannans, enhanced intestinal characteristics at the bacteriological and morphological level [19]. The main commercial sources of galactomannans are guar gum (*Cyamopsistetragonolobo*, with a mannose/galactose ratio 2:1), tara gum (*Caesalpiniaspinosa,* 3:1) and locust bean gum (*Ceratoniasiliqua*, 3.5:1) [20]. However, another unconventional source of galactomannan that could be found in Mediterranean countries is *Trigonella foenum-graecum*, known as fenugreek. Although there is no information on the use of galactomannans of this origin in animals, they have exhibited prebiotic activity in in vitro models, as well as suitability with probiotics in a symbiotic combination [21]. These results could indicate that galactomannan from fenugreek seeds could be a good candidate as a prebiotic in rabbits. Therefore, the aim of this study initially was to characterise the fenugreek seed gum (FGS) obtained by extraction, and then determine whether this extract may be a potential prebiotic in rabbit diets enriched with soluble or insoluble fibre, through an in vitro model of an initial enzymatic digestion and subsequent fermentation with rabbit caecal inoculum.

## 2. Materials and Methods

Animal housing, husbandry and slaughtering conditions followed the current recommendations on principles of ethical care and protection of animals used for experimental purposes in the European Union (2003) and all trials were subjected to approval by the Animal Protocol Review Committee of the Polytechnic University of Valencia (research code: 2018/VSC/PEA/0116). The experiment was also carried out following the recommendations for applied nutrition research in rabbits described by the European Group on Rabbit Nutrition [22].

### 2.1. Fenugreek Seed Gum Characterisation

Fenugreek seeds were bought from the weekly market of the Mateur region in the North of Tunisia. As described in Reference [23], from a batch of ground fenugreek seeds, three subsamples (200 g each) were used for gum extraction. Each extraction consisted of twice defatting in 200 mL hexane-isopropanol (3/2, *v*/*v*), washing with 150 mL acetone, solubilising in 2 L of distilled water for 24 h, precipitating with ethanol (1 L), freeze-drying and grinding to pass 1 mm diameter. For an initial characterisation, chemical composition, the physical properties and fermentation pattern of the FSG obtained were determined. Regarding its chemical composition, the FSG was subjected to a preliminary qualitative phytochemical screening test for the detection of steroids, terpenoids, flavonoids, tannins, alkaloids, coumarins, saponins, free amino acids and reducing sugar in its composition. Moreover, the FSG was analysed to determine the content in dry matter (DM), ashes, crude protein (CP), total sugars, sugars composition and amino acid composition. Neutral detergent fibre (NDF) content of fenugreek gum was also determined, but gels’ production, due to its interaction with sugars, makes its determination ambiguous. The chemical composition methodology used is described in Section 2.3. As regards its physical properties, the bulk density of the FSG was determined using the method described by Narayana and Narasinga-Rao [24]. A calibrated tube was weighed and filled with a sample to 5 mL by constantly tapping until there was no further change in volume. Bulk density was then calculated as the weight determined by difference per unit volume of the sample. The solubility of the extract in cold water (10–15 °C) was also determined, adding 50 mL of cold water to 0.50 g of the FSG, as described by Torio et al. [25]. Finally, the FSG fermentation pattern was determined using the gas production technique. The rabbit caecal inoculum was obtained in the morning at the laboratory of the Higher School of Agriculture of Mateur (Cartago, Tunisia). Two New Zealand White adult rabbits were fasted overnight (with free access to freshwater) before slaughter at early morning. Once the whole gastro-intestinal tract had been isolated, caecal contents were collected, mixed and stirred for few minutes under CO_2_ bubbling. Finally, 100 mL of the mixture was homogenised under 1 L of solution A (including 10 g/L KH_2_PO_4_, 0.5 g/L Mg_2_SO_4_·7H_2_O, 0.5 g/L NaCl and 0.1 g/L CaCl_2_·2H_2_O) and 8 mL of solution B (including 15 g/L Na_2_CO_3_ and 0.25 g Cysteine- HCl) and the inoculum solution obtained was filtered through six layers of cheese cloth [26]. Propylene sterile syringes of 60 mL were used for the incubation. Twenty mL of inoculum solution was included in each syringe, with 200 mg of FSG or without for the blank. As a result of the low level of protein FSG, the incubations were performed without and with the addition of 0.5 g/L of urea in the buffer (solutions A + B) to facilitate fermentation. A total of 22 syringes were used in this trial, three replicates for each of the three FSG extractions at two levels of urea addition, as well as another four for the blanks (inoculum solution without substrate). Syringes were quickly placed in an oven and incubated at 39 °C for 28 h. Gas production was recorded at 0.5, 1, 3, 5, 7, 9, 11, 17, 22, 24, 26 and 28 h post inoculation. Gas production values at each measurement time were corrected for gas produced at this time by the corresponding blanks. Gas production data were fitted to the logistic model described by Schofield et al. [27]:(1)$$G=\frac{V_{f}}{\left(1+e^{\left(2-4k\left(t-Lag\right)\right)}\right)}$$

In this equation, *G* (mL) is the asymptotic amount of gas produced, *k* (h^−1^) is the fractional rate of gas production, *Lag* is the delay in the onset of gas production and *t* is the incubation time. Maximum gas production rate (µm, mL/h) and time at which µm is achieved (*t_µm_*) were also calculated as:(2)$$\mu_{m}=k\times V_{f}\;\mathrm{and}\;t_{\mu m}=Lag+\left(V_{f}/\left(2\times \mu_{m}\right)\right)$$

### 2.2. In Vitro Digestive Behaviour

As a potential prebiotic for rabbits, the in vitro digestive behaviour of FSG was evaluated, both in pure form and when included up to 20 g/kg in diets rich in soluble or insoluble fibre. From a basal diet, two experimental diets were formulated to differ in their content in soluble and insoluble fibre provided by directly adding 100 g/kg of beet pulp (diet SF) or 100 g/kg of grape seeds (diet IF) to the basal diet respectively, but maintaining minerals and premix supply. The diets were formulated to include CP and acid detergent fibre (ADF) levels close to the current recommendations [28]. Ingredients and chemical composition of the diets are given in Table 1.

From these experimental diets, 11 substrates were prepared: SF0, SF5, SF10, SF15 and SF20, SF diet including 0, 5, 10, 15 and 20 g/kg of FSG, respectively. IF0, IF5, IF10, IF15 and IF20, IF diet including 0, 5, 10, 15 and 20 g/kg of FSG respectively, and pure FSG. In a first step, the behaviour of the 11 substrates under an in vitro simulation of stomach and intestinal enzymatic digestion was evaluated. The enzymatic digestion was done according to the multi-enzymatic method of Ramos and Carabaño [29]. Using a Daisy incubator (Ankom Technology Corp., Macedon, NY, USA), three replicates of 0.5 g of each substrate were weighed in Ankom bags, placed in the Daisy incubator jar at 40 °C with 25 mL/bag of the buffer solution A (0.1M, a pH = 6; composed of 0.99 g of sodium phosphate dihydrate (Na_2_HPO_4_·2H_2_O) and 14.72 g of sodium phosphate monobasic dihydrate (NaH_2_PO_4_·2H_2_O) in 1 L of deionised water) with 10 mL/bag of 0.2M hydrochloride solution, adjusting the pH value to 2. At this level, pepsin solution (0.025 g/sample and mL of HCl 0.2M) was added to the jar and maintained for 2 h. Then, 10 mL/bag of the buffer B (0.2M, pH = 6.8; composed of 9.65 g of sodium phosphate dihydrate (Na_2_HPO_4_·2H_2_O) and 22.74 g of sodium phosphate monobasic dihydrate (NaH_2_PO_4_·2H_2_O) in 1 L of deionised water), 5 mL/bag of sodium hydroxide (6M) and 100 mg of pancreatin/bag were added to the jar. Agitation was maintained for 3 h and 30 min at 40 °C. At the end of the enzymatic digestion, the Ankom bags were washed over 3 times with distilled water at 40 °C and then dried at 105 °C during 12 h in the stove. Finally, the bags were weighed to determine the indigestible fraction after pepsin and pancreatin enzymatic digestion. In a second step, the behaviour of the 11 substrates was evaluated, obtained during the enzymatic digestion under an in vitro simulation of caecal fermentation. The in vitro fermentation procedure was based on the method described by Fernández-Carmona et al. [30]. Five young rabbits from the LP line (Universitat Politècnica de València, Valencia, Spain) were weaned at 28 days of age and received a commercial rabbit diet until slaughter age (56 days of age). Rabbits had ad libitum access to feeders and fresh water during the trial without any antibiotic treatment. After slaughter, caecal content was collected and mixed in an anaerobic environment. Under constant CO_2_ bubbling, 400 g of the caecal mixture was homogenised with 340 mL of artificial saliva NaHCO_3_ (8 g), K_2_HPO_4_ (4 g), (NH_4_)2HPO_4_ (0.5 g), NaCl (1.5 g) and MgSO_4_·7H_2_O (0.5 g) per L of deionised water and filtrated through a cellulose filter. The filtrate was then centrifuged at 6000 rpm and 38 °C for 10 min, and the supernatant was homogenised with another 500 mL of the artificial saliva under CO_2_ bubbling. Finally, the inoculum obtained was diluted up to 4 times the initial weight of caecal content by adding 1.6 L of artificial saliva. The inoculum solution was prepared on the day of fermentation, under constant CO_2_ at 38 °C, and the pH was adjusted to 6.9. A 0.7 g amount of each substrate was carefully weighed in a 120 mL bottle with 50 mL of inoculum. The bottle was sealed and incubated for 28 h in the stove with frequent gentle agitation. Two bottles were used for each substrate and other two for the blank (inoculum without any substrate). At the end of the incubation time, all bottles were placed in an ice bath to stop fermentation. Gas pressure was measured before opening the bottles. After opening, pH was determined, and one bottle was used to determine the unfermented fraction and the second bottle for total volatile fatty acids (VFA), VFA profile and N-NH_3_. One hundred µL of 0.35 M H_3_PO_4_ (including 4-methyl valeric acid) or 3 mL of 0.35 M H_2_SO_4_ were added to 900 µL of the fermentation product for later determination of VFA and N-NH_3_. This in vitro simulation was repeated 3 times on different days.

### 2.3. Chemical Analysis

A preliminary qualitative phytochemical screening was carried out. Using an aqueous extract of the FSG obtained, the presence of steroids [31], terpenoids [32], flavonoids [33], tannins [34], alkaloids [35], saponins [36], coumarins [37] and reducing sugars (Benedict’s test [38]) were determined. FSG was analysed for DM, ash, CP and sugar content, as well as for amino acids and monomers profile of their protein and sugars, respectively. The diets were analysed for DM, ash, CP, NDF, acid detergent fibre (ADF) and acid detergent lignin (ADL), as well as for total dietary fibre (TDF) and soluble fibre in IF and SF diets. The indigestible fraction after pepsin and pancreatin digestion was analysed for DM, ash, CP, NDF, ADF, ADL and starch. Finally, the bottle content after fermentation was analysed for gas production, DM, pH, total VFA, VFA profile and N-NH_3_. Samples were analysed according to the methods of Association of Official Analytical Chemists (AOAC) [39], 934.01 for DM, 942.05 for ash and 990.03 for CP. Starch content was determined according to Batey [40], by a two-step enzymatic procedure with solubilisation and hydrolysis to maltodextrins with thermostable α-amylase followed by complete hydrolysis with amyloglucosidase (both enzymes from Sigma-Aldrich, Steinheim, Germany), and the resulting D-glucose being measured by the hexokinase/glucose-6 phosphate dehydrogenase/NADP system (kit D-glucose-HK Megazyme Int. Ireland Ltd., Wicklow, Ireland). The NDF, ADF and ADL were analysed sequentially [41] by the AOAC [39] procedure 973.18 and Robertson and Van Soest [42] with a thermostable α-amylase pre-treatment and expressed exclusive of residual ash, using a nylon filter bag system (Ankom Technology Corp., Macedon, NY, USA). The TDF content was determined by a gravimetric-enzymatic method, the procedure 991.43 of the Van Soest et al. [43], with α-amylase, protease and amyloglucosidase treatments (Megazyme TDF R.30.K-TDFR-100A/200A), correcting for ash and CP (method 985.29). Hemicellulose, cellulose and soluble fibre contents were calculated by difference as NDF − ADF, ADF − ADL and TDF − NDF, respectively. The amino acid content of FSG protein was determined after acid hydrolysis with HCl 6N at 110 °C for 23 h, as previously described by Bosch et al. [44], using Waters (Milford, MA, USA) HPLC system consisting of two pumps (Model 515, Waters, Milford, MA, USA), an auto-sampler (Mod. 717, Waters, Milford, MA, USA), a fluorescence detector (Mod. 474, Waters, Milford, MA, USA) and a temperature control module. Aminobutyric acid was added as an internal standard after hydroxylation. The amino acids were derivatised with AQC (6-aminoquinolyl-N-hydroxysuccinimidyl carbamate) and separated with a C-18 reverse-phase column Waters AQC Tag (150 × 3.9 mm). Methionine and cysteine were determined separately as methionine sulfone and cysteic acid respectively, after performic acid oxidation followed by acid hydrolysis. To determine total VFA concentration and VFA profile, samples were centrifuged for 5 min during 15,000 rpm, and 300 µL from the supernatant was filtrated through a syringe filter with porosity of 0.22 µ. Then, 200 µL of the filtrate was recuperated in an injection vial. Two microlitres from each sample were injected into the gas chromatograph (Fisons 8000 series, Milan, Italy) equipped with an AS800 automatic injector. The column used was a BD-FFAP 30 m × 0.25 mm × 0.25 mm. Injector and detector temperatures were maintained at 220 and 225 °C, respectively. For N-NH_3_, 3 mL of sulphuric acid (0.35 M) was mixed with 1 g of fermentation product. Ammonia concentration was determined according to procedure 984.13 of the AOAC [39]. The VFA and N-NH_3_ concentrations were expressed as mmol L-1 of the liquid phase of fermentation product.

### 2.4. Statistical Analysis

Digested, fermented and neither digested nor fermented FSG fractions were calculated in DM basis. The indigestible and unfermented fractions were calculated concerning the DM of the initial FSG and the indigestible fraction, respectively. Data on gas production kinetics were analysed using a general linear model (GLM) procedure of SAS [45], with a model including the urea addition (0 and 0.5 g/L) as a fixed effect. Data corresponding to the content and composition of the indigestible and unfermented fractions were analysed using a GLM procedure of SAS (SAS Institute Inc., Cary, NC, USA) [45] with a model including the substrate (SF0, SF5, SF10, SF15, SF20, IF0, IF5, IF10, IF15, IF20 and FSG) as a fixed effect. Different contrasts were computed to test the significance of the differences between diets (SF − IF = SF0 + SF5 + SF10 + SF20 – IF0 – IF5 – IF10 – IF20)/5), of the inclusion of pure FSG (0 versus 1000 = (SF0 + IF0)/2 – FSG), and of the linear effect of the FSG in all diets, SF diets and IF diets.

## 3. Results

FSG extracted was a white powder, with low humidity (88 g/kg) and ashes (13 g/kg DM), and a moderate protein content (223 g/kg DM), whose profile is shown in Table 2.

As expected, FSG was rich in total sugars (630 g/kg DM), consisting entirely of galactose and mannose, in a ratio of 1:1. In contrast, it can be seen that there is 134 g/kg that should correspond to some type of fibrous fraction. However, when trying to determine these fibrous fractions sequentially, the gels formed interacted with the sugars, rendering such determination ambiguous, so it was discarded. The preliminary qualitative phytochemical screening tests reported that the FSG contained traces of flavonoids, tannins and coumarins, while it had no steroids, terpenoids, alkaloids, saponins or reducing sugars. FSG had a bulk density of 0.2 ± 0.04 g/mL and solubility in cold water of 0.0042 g/mL.

Table 3 shows the gas production kinetics of FSG after 28 h in vitro fermentation, when adjusted to a logistic model. The inclusion of urea in the buffer led to a decrease in *k* (−0.03 ± 0.01 h-1; *p* < 0.05), *Lag* (−2.04 ± 0.34 h; *p* < 0.01), *µ_m_* (−0.60 ± 0.18 h; *p* < 0.05) and *t_µm_* values (−1.35 ± 0.18 h; *p* < 0.01).

Table 4 shows the effect of FSG inclusion in the experimental diets on the amount and composition of the indigestible fraction after in vitro enzymatic digestion with pepsin and pancreatin. When the experimental diets are compared, the indigestible fraction of the SF diets had less NDF, ADF and ADL (from 47 to 75 g/kg; *p* < 0.001), but slightly more CP (7 g/kg; *p* < 0.001) than the indigestible fraction of the IF diets. Pure FSG was affected very little by in vitro enzymatic digestion, only 145 g/kg of the FSG was dissolved during the enzymatic digestion compared to 465 g/kg of experimental diets (*p* < 0.001; Figure 1, dark grey bars).

The indigested fraction of pure FSG was richer in crude protein (+55 ± 4 g/kg; *p* < 0.001) and poorer in starch (−39 ± 6 g/kg; *p* < 0.001) than the indigestible fraction of the experimental diets. FSG inclusion up to 20 g/kg significantly increased the indigestible fraction of the diets, mainly due to its effect on SF diets (+1.4 g/kg per each g/kg of FSG added; *p* < 0.001). In addition, FSG inclusion linearly increased the starch and CP content of the indigestible fraction but decreased the NDF and cellulose (+0.4, +0.6, −0.8 and −0.6 g/kg per each g/kg of FSG added, respectively; *p* < 0.001). However, this modification of indigestible fraction associated with linear inclusion of FSG was only observed in FS diets (+0.7, +0.9, −1.6 and −0.9 g/kg per each g/kg of FSG added; *p* < 0.001), being unaffected in IF diets (Figure 2).

Finally, Table 5 shows the effect of FSG inclusion in the experimental diets on the unfermented fraction and some fermentative traits after an in vitro fermentation with caecal inoculum of the indigestible fraction previously obtained.

When the experimental diets are compared, indigestible fraction of SF diets is more fermented (+83 g/kg; *p* < 0.001) with a greater gas production (+0.15 mbar; *p* < 0.001) and total VFA (+5.7 mmol/L; *p* < 0.001), whose profile contained a lower proportion of butyric, isobutyric and isovaleric acids (–2.3, −0.1 and −0.1 percentage points; *p* < 0.05). In addition, pH and N-NH_3_ content of the final fermentation product was also lower with the SF diet than with IF (−0.12 points and −15.4 mmol/L, respectively; *p* < 0.001). Indigestible fraction of pure FSG was almost completely used during in vitro fermentation, at 984 g/kg of the indigestible fraction. As shown in Figure 1 (clear green bars), FSG 841 g/kg of the initial FSG was fermented, clearly higher (*p* < 0.001) than the fermented fraction of SF diets (on average, 226 g/kg), which was significantly higher than that of IF diets (on average, 183 g/kg). Comparing it to diets without FSG, pure FSG reduced the pH and concentration of N-NH_3_ after fermentation (*p* < 0.001). It also produced a clear increase in gas production and total VFA content (*p* < 0.001), whose profile was enriched in propionic, isobutyric and isovaleric, and depleted in valeric and caproic acids. In addition, linear dietary FSG inclusion increased the fermented fraction and the relative proportion of valeric acid (+0.6 g/kg and +0.01 percentage points per each g/kg of FSG added; *p* < 0.05). However, this modification of relative proportion of valeric acid associated with linear inclusion of FSG was only observed in FS diets (+0.01 percentage points per each g/kg of FSG added; *p* < 0.05), being unaffected in IF diets.

## 4. Discussion

In the current research work, our main purpose was to evaluate the potential prebiotic effect of FSG, both pure and when included up to 20 g/kg in diets rich in soluble or insoluble fibre, as its prebiotic potential could be different depending on the nature of the main dietary fibre. For this reason, in the present work, we initially characterised the FSG extracted, and subsequently, using a enzymatic digestion and fermentation simulation, evaluated the digestive and fermentative behaviour of the FSG pure or when added to diets differing in soluble or insoluble fibre. Regarding FSG chemical composition, the preliminary analysis of phytochemicals seems to indicate that FSG only contains traces of some phenolic compounds, and absence of alkaloids, steroids, terpenoids and reducing sugars. Therefore, a priori, the FSG extract obtained would not present anti-nutritive factors or a level of toxicity that negatively affects rabbits’ performance when included in their diets. Accordingly, Deshpande et al. [46] evaluated the possible toxicity in rats of an extract of fenugreek seeds, rich in galactomannan of low molecular weight, and observed that their inclusion up to 1000 mg/kg/day did not produce negative effects on ingestion, haematology, biochemistry and histology of animals, even in mutagenicity, and can be considered safe. FSG is mostly composed of galactomannan (galactose and mannose: 630 g/kg). Similar results (635 g/kg) were found by Majeed et al. [21] for a commercial galactomannan from fenugreek seeds. The purity reported for other galactomannans obtained from carob and mesquite seeds was greater, above 940 g/kg [47,48]. These purity values were higher than those obtained for FSG due to two possible reasons: the separation of the endosperm from the seeds and the purification processes, as the extraction procedure is almost the same. In the case of fenugreek seeds, endosperm cannot be separated, and the purification procedure is expensive. A further improvement in this process could improve the potential of the results obtained with the FSG. In our case, the FSG extract obtained had a moderate content in CP (223 g/kg DM), with an amino acid profile quite similar to that obtained for fenugreek seed protein [49].

Gas production kinetics can reflect the extent to which substrates are used by caecal microbiota [50]. FSG was highly fermented by rabbits’ caecal microflora because of its high carbohydrate content, which is positively correlated with the maximum cumulative volume of gas production [51]. The fermentative behaviour of FSG was similar to that shown by other soluble fibre sources in rabbits [52,53], having a maximum gas production rate (3.1 to 3.7 mL/h) between that of sugar beet pulp (2.9 mL/h) and that of pectin (6.0 to 8.0 mL/h). The high level of microbial fermentation of FSG might imply high energy losses, which may require a nitrogen source supplementation. This fact could explain the increase in the fractional and maximum gas production rates with the urea addition. As expected, diets enriched in SF (sugar beet pulp) showed a higher degree of enzymatic digestion and fermentation in vitro than diets enriched in IF (grape seeds). Trocino et al. [6], in a meta-analysis, concluded that an increase in SF to TDF ratio increases both digested and fermented TDF. TDF digestibility seems to increase because of the high digestibility of SF per se, and the low lignification level of the raw materials rich in SF. The experimental diets did not differ too much in NDF, but ADF and ADL were clearly lower in SF diets. On the other hand, SF stimulates the growth of fibrolytic microbiota in the gut [54], and consequently, fibre fermentation, reduction of caecal pH and production of VFA, usually promoting acetate with respect to butyrate, as in the present work [55].

Gibson and Roberfroid [56] defined a prebiotic as “a non-digestible fibre in the upper gastro-intestinal tract which can enhance the growth or activity of advantageous bacteria of large bowel by acting as a substrate for them”. Later, Roberfroid [57] added that a prebiotic is “a selectively fermented ingredient that allows specific changes both in the composition and/or activity in the gastro-intestinal microflora that confers wellbeing and health benefits for the host”. This means that to be prebiotic, a feed ingredient should have a composition that would allow it to be not only non-digestible in the gastro-intestinal tract but also highly fermentable in the caecum and the colon (as in the case of rabbits). These characteristics have been clearly fulfilled by the pure FSG, as it was little affected by in vitro enzymatic digestion and almost completely used up during in vitro fermentation. These results agree with those obtained by Majeed et al. [21], who observed that when fenugreek seed galactomannan was subjected to gastric and subsequently pancreatic digestion, the release of reducing sugars was practically nil. In addition, Majeed et al. [21] also observed that *Bacillus coagulans* could grow in a medium that only included fenugreek seeds galactomannan as nutrient (0.5% *w*/*v*), showing a better count compared to that observed with other prebiotics such as fructooligosaccharides, confirming FSG’s potential as a prebiotic. Therefore, it seems that FSG promotes substrate fermentation by the rabbit caecal microbiota, reducing the caecal pH, increasing the production of total VFA and consuming the N-NH_3_ available in the medium by the action of the fibrolytic bacteria [58]. Regarding the fermentation profile, the results seem to show that pure FSG increases the proportion of propionate, iso-butyrate and isovalerate, but reduces caproate when compared to the profile obtained with experimental diets, while the inclusion of soluble fibre from beet pulp promotes the proportion of butyrate and isovalerate regarding the inclusion of lignin from grape seeds. Marounek et al. [59] observed that the molar proportions of propionate and caproate were promoted when rabbit caecal cultures were supplied with inulin and starch, respectively. Thus, the inclusion of prebiotic additives, such as inulin or fenugreek gum, seems to promote the proportion of propionate, while feeds with higher starch content would lead to a higher proportion of caproate. However, the inclusion of beet pulp, with a proven prebiotic effect in rabbits, leads to a different profile. This may be due to the fact that, during the enzymatic digestion, diets rich in beet pulp produce an indigestible fraction richer in highly fermented hemicellulose and cellulose, which promote butyrate production when they reach the caecum [60]. In the case of isobutyrate, isovalerate and valerate, they are not produced by fermentation of carbohydrates but from the degradation of the amino acids, valine, leucine and proline [61]. The higher proportion of iso-butyrate and iso-valerate during fermentation of pure FSG may be related to its relatively high protein content (223 g/kg DM), which was also rich in leucine and had a non-negligible valine content. Linear increase of FSG inclusion up to 20 g/kg reduced the digestive use of the diets enriched in soluble fibre during the enzymatic digestion. As the only dietary change in SF diets was the inclusion of 100 g/kg of sugar beet pulp, it seems that the inclusion of FSG as a prebiotic could be altering the solubilisation of the sugar beet pulp during the enzymatic digestion with pepsin and pancreatin. On the other hand, in the indigestible fraction obtained with SF diets, protein and starch proportions were increased, but the fibre fractions proportions were reduced as the FSG increased (e.g., −1.6 g/kg of NDF per each g/kg of FSG added). Other authors also observed that an increase of soluble fibre increased the faecal and ileal digestibility of the main fibre fractions of the diet (on average, 0.6 to 0.8 g/kg of soluble NDF per each g/kg of soluble fibre added; Trocino et al. [54] and Gómez-Conde et al. [55]). However, this effect has not been observed when other potential prebiotics have been included in the diet of growing rabbits (e.g., inulin; [62]). These results may be related to some of the physical aspects of gums such as FSG. Some studies [63,64] have reported that the dietary inclusion of gums reduced starch and protein hydrolysis during simulated gastric and intestinal digestion. Gums may be acting by forming a barrier layer around the feed or increasing digesta viscosity that could restrict the access of the enzymes to the granules of starch and protein of the digesta, reducing its solubilisation, especially in diets enriched with sugar beet pulp [6]. Finally, although FSG was almost totally fermented by the rabbit caecal bacteria, its inclusion up to 20 g/kg in the diet only slightly increased the in vitro fermentation of the diets where they were included (probably related to its own disappearance), without relevant changes in the in vitro fermentation profile observed in the caecum content of the rabbits, with only a slight increase in the content of valerate.

## 5. Conclusions

In the present work, we have tried to evaluate the potential of an extract of FSG, rich in galactomannan, as a functional product with prebiotic activity for growing rabbits. In this way, FSG satisfies two essential conditions of the prebiotic effect: it shows resistance to in vitro enzymatic digestion and was totally fermented in vitro by caecal bacteria. Consequently, pure FSG promoted the caecal fermentation as expected for a prebiotic, reducing caecal pH and N-NH_3_ content, increasing total VFA production and modulating VFA profile. However, FSG linear inclusion up to 20 g/kg in growing feeds led to a slight reduction in the solubility of some nutrients during the in vitro enzymatic digestion, and although FSG increased the fermentation fraction of the diets, no relevant changes in the fermentation profile were observed. For this reason, it is necessary to perform in vivo studies with different sources of fibre to truly determine the prebiotic efficiency of FSG rich in galactomannan.

## Figures and Tables

**Figure 1 animals-10-01041-f001:**
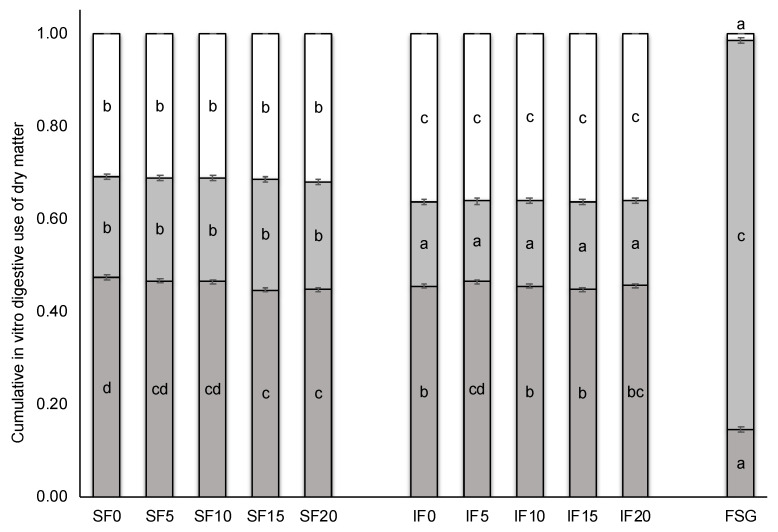
Effect of fenugreek seed gum (FSG) level of inclusion in the experimental diets (SF and IF) on their cumulative in vitro digestive use of dry matter (DM): digested faction (dark grey), fermented fraction (light grey) and fraction neither digested nor fermented (white). FSG, pure fenugreek seed gum; SF0, SF5, SF10, SF15 and SF20, SF diet with 0, 5, 10, 15 and 20 g of FSG included per kg of feed; IF0, IF5, IF10, IF15 and IF20, IF diet with 0, 5, 10, 15 and 20 g of FSG included per kg of feed. ^a,b,c,d^ Bars for each fraction not sharing a letter were significantly different at *p* < 0.05.

**Figure 2 animals-10-01041-f002:**
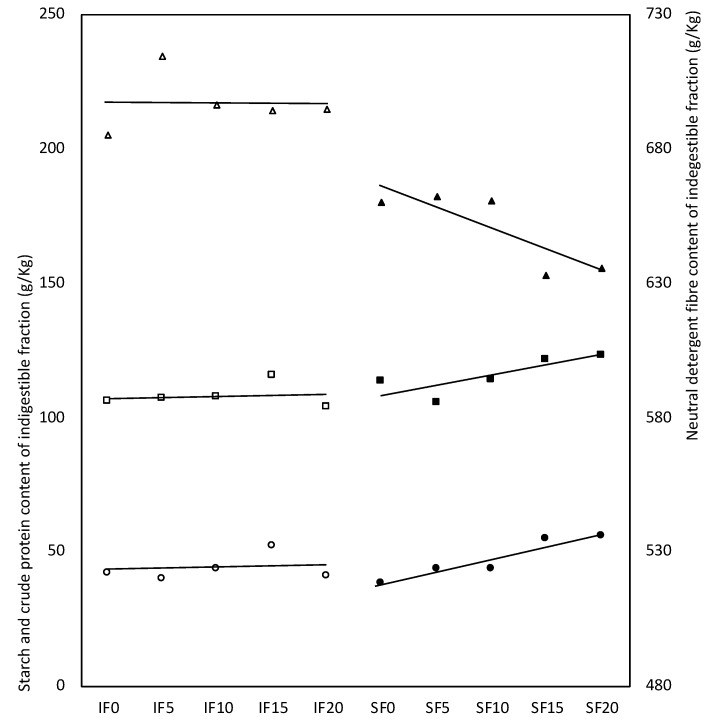
Effect of fenugreek seed gum (FSG) level of inclusion in the experimental diets (SF and IF) on the ◯ starch, ☐ crude protein and △ neutral detergent fibre content of indigestible fraction after in vitro incubation with pepsin and pancreatin (symbols without filling in IF diets and with filling in SF diets). SF0, SF5, SF10, SF15 and SF20, SF diet with 0, 5, 10, 15 and 20 g of FSG included per kg of feed; IF0, IF5, IF10, IF15 and IF20, IF diet with 0, 5, 10, 15 and 20 g of FSG included per kg of feed.

**Table 1 animals-10-01041-t001:** Ingredients (g/kg) and chemical composition (g/kg DM) of the diets.

Ingredients and Chemical Composition	Basal Diet ^a^	Experimental Diets	
		SF	IF
Ingredients			
Wheat bran	247	221	221
Sunflower meal	212	190	190
Beet pulp	108	197	97
Grape seeds	0	0	100
Alfalfa hay	100	90	90
Wheat straw	80	72	72
Corn flour	80	72	72
Corn germ cake	49	44	44
Molasses	40	36	36
Rice bran	30	27	27
Corn grain	25	22	22
Calcium carbonate	11	11	11
Sodium chloride	3	3	3
Premix ^b^	15	15	15
Chemical composition			
Dry matter (DM, g/kg)	898	896	895
Ash	70.0	70.1	66.4
Crude protein	176	168	169
Neutral detergent fibre	397	401	415
Acid detergent fibre	197	204	231
Lignin acid detergent	29.7	27.7	66.1
Total dietary fibre	-	489	474
Soluble fibre	-	88.1	59.0

^a^ Commercial rabbit diet provided to the young rabbits after weaning. ^b^ Supplied per kg of feed: Biolys, 1.2 g (0.66 g L-Lysine); Calcium chloride Hill, 0.15 g; Lignobond, 10 g (lignosulphonate feed binder); Vitamin/trace element premix NL-310-Vita, 3.0 g (vitamin A, 8375 IU; vitamin D3, 750 IU; vitamin E, 20 mg; Vitamin K3, 1 mg; vitamin B1, 1 mg; vitamin B2, 2 mg; vitamin B6, 1 mg; nicotinic acid, 20 mg; choline chloride, 250 mg; magnesium, 290 mg; manganese, 20 mg; zinc, 60 mg; iodine, 1.25 mg; iron, 26 mg; copper, 10 mg; cobalt, 0.7 mg; butyl hydroxylanysole and ethoxiquin mixture, 4 mg); Luctanox, 0.2 g (antioxidant); Luctarom: 0.5 g (flavouring). Not determined. SF, soluble fibre; IF, insoluble fibre.

**Table 2 animals-10-01041-t002:** Chemical composition of fenugreek gum.

Chemical Composition	(g/kg DM)
Dry matter (DM) (g/kg)	913
Ashes	12.7
Crude protein (CP)	223
Sugars	630
Sugar composition (g/kg sugars):	
Galactose	515
Mannose	485
Fructose	<1
Glucose	<1
Lactose	<1
Maltotriose	<1
Galactose/mannose ratio	1.1
Amino acid composition (g/kg CP):	
Alanine	25.9
Arginine	76.3
Aspartate	107
Cysteine	13.4
Glutamate	172
Glycine	27.9
Histidine	22.3
Isoleucine	47.9
Leucine	64.1
Lysine	57.9
Methionine	9.10
Phenylalanine	33.1
Proline	32.9
Serine	39.4
Threonine	23.1
Tyrosine	15.2
Valine	32.1

Neutral detergent fibre content of fenugreek gum was also determined (730.6 g/kg DM), but gel production during analysis made its determination uncertain.

**Table 3 animals-10-01041-t003:** Effect of urea addition in the buffer on gas production kinetics of fenugreek seed gum in 28 h in vitro incubations with caecal inoculum (n = 9 for each level).

Gas Production Kinetics ^a^	Urea Addition (g/mL)			
	0	0.5	SEM	*p*-Value
*V_f_* (mL)	24.46	24.92	1.597	0.8545
*k* (h^−1^)	0.152	0.126	0.006	0.0318
*Lag* (h)	6.118	4.077	0.245	0.0047
*µ_m_* (mL/h)	3.722	3.119	0.130	0.0340
*t_µm_* (h)	9.411	8.063	0.127	0.0019

^a^*V_f_*, asymptotic gas production; *k*, fractional rate of gas production; *Lag*, initial delay in the onset of gas production; *μ_m_*, maximum gas production rate; *t_μm_*, time when *μ_m_* is reached (logistic model described by Schofield et al. [27]). SEM, standard error of the mean.

**Table 4 animals-10-01041-t004:** Effect of fenugreek seed gum (FSG) inclusion in the experimental diets (SF and FI) on the amount and composition of the indigestible fraction after an in vitro enzymatic digestion with pepsin and pancreatin (least square means ± standard error).

Traits	Mean	Contrasts ^1^		Linear Effect of FSG Inclusion		
		SF Versus IF	0 Versus 1000	All Diets	SF Diets	IF Diets
Indigestible fraction (g/kg)	560.8	−4.05 ± 2.68	−319.5 ± 6.1 ^***^	0.86 ± 0.02 ^***^	1.43 ± 0.27 ^***^	0.29 ± 0.27
Indigestible fraction composition (g/kg):						
Crude protein	116.5	7.45 ± 1.60 ^***^	−54.5 ± 3.1 ^***^	0.39 ± 0.11 ^**^	0.70 ± 0.16 ^**^	0.01 ± 0.16
Neutral detergent fibre	673.6	−46.70 ± 2.22 ^***^		−0.79 ± 0.15 ^***^	−1.57 ± 0.22 ^***^	−0.01 ± 0.22
Acid detergent fibre	381.9	−64.27 ± 3.96 ^***^		−0.41 ± 0.28	−1.08 ± 0.40 ^*^	0.25 ± 0.40
Lignin acid detergent	101.2	−74.75 ± 3.30 ^***^		0.13 ± 0.23	−0.16 ± 0.33	0.42 ± 0.33
Hemicellulose	291.7	17.57 ± 3.36 ^***^		−0.38 ± 0.24	−0.49 ± 0.34	−0.26 ± 0.34
Cellulose	280.8	10.48 ± 2.03 ^***^		−0.55 ± 0.14 ^**^	−0.92 ± 0.20 ^**^	−0.17 ± 0.20
Starch	41.5	−3.49 ± 2.20	39.1 ± 4.3 ^***^	0.56 ± 0.16 ^**^	0.93 ± 0.22 ^***^	0.20 ± 0.22

^1^ SF, soluble fibre diets; IF, insoluble fibre diets; FSG 0 versus 1000, FSG inclusion level (g/kg dry matter): 0 ((SF0 + IF0)/2) and 1000 (pure FSG). Significance: * *p* < 0.05; ** *p* < 0.01; *** *p* < 0.001.

**Table 5 animals-10-01041-t005:** Effect of fenugreek seed gum (FSG) inclusion in the experimental diets (SF and FI) on some fermentation traits after an in vitro fermentation with caecal inoculum of the indigestible fraction obtained from the in vitro enzymatic digestion with pepsin and pancreatin (least square means ± standard error).

Traits	Mean	Contrasts ^1^		Linear Effect of FSG Inclusion		
		SF versus IF	0 versus 1000	All Diets	SF Diets	IF Diets
Unfermented fraction (g/kg)	566.9	−83.1 ± 3.7 ^***^	610.8 ± 7.1 ^***^	−0.6 ± 0.3 ^*^	−0.7 ± 0.4	−0.6 ± 0.4
Gas pressure (mbar)	0.62	0.15 ± 0.02 ^***^	−0.61 ± 0.03 ^***^	0.00 ± 0.00	0.00 ± 0.00	0.00 ± 0.00
pH	6.18	−0.12 ± 0.02 ^***^	0.68 ± 0.03 ^***^	0.00 ± 0.00	0.00 ± 0.00	0.00 ± 0.00
Total VFA (mmol/L)	19.49	5.72 ± 1.46 ^***^	−27.32 ± 3.31 ^***^	0.03 ± 0.10	−0.04 ± 0.15	0.11 ± 0.15
VFA profile (%):						
Acetate	63.25	1.94 ± 1.64	4.28 ± 3.72	−0.05 ± 0.12	−0.11 ± 0.16	0.01 ± 0.16
Propionate	11.63	0.21 ± 0.38	−3.08 ± 0.87 ^**^	0.05 ± 0.03	0.05 ± 0.04	0.04 ± 0.04
Butyrate	21.84	−2.29 ± 1.04 ^*^	−2.12 ± 2.35	−0.02 ± 0.07	0.01 ± 0.10	−0.06 ± 0.10
Iso-butyrate	0.37	−0.07 ± 0.03 ^*^	−0.15 ± 0.06 ^*^	0.00 ± 0.00	0.00 ± 0.01	0.00 ± 0.00
Valerate	0.70	0.02 ± 0.04	0.26 ± 0.09 ^*^	0.01 ± 0.00 ^*^	0.01 ± 0.00 ^*^	0.00 ± 0.00
Isovalerate	0.31	−0.09 ± 0.04 ^*^	−0.49 ± 0.09 ^***^	0.00 ± 0.00	0.01 ± 0.00	0.00 ± 0.00
Caproate	1.83	0.27 ± 0.17	1.23 ± 0.39 ^**^	0.01 ± 0.01	0.03 ± 0.02	−0.00 ± 0.02
Heptanoate	0.09	0.01 ± 0.02	0.06 ± 0.04	0.00 ± 0.00	0.00 ± 0.00	0.00 ± 0.00
N-NH_3_ (mmol/L)	28.89	−15.39 ± 3.09 ^***^	65.15 ± 5.89 ^***^	0.33 ± 0.22	0.67 ± 0.33	−0.00 ± 0.29

^1^ Mean value obtained with respect to the blank (caecal inoculum without any substrate addition); soluble fibre diets; IF, insoluble fibre diets; 0 versus 1000, FSG inclusion level (g/kg dry matter): 0 ((SF0 + IF0)/2) and 1000 (pure FSG). VFA, volatile fatty acids. Significance: * *p* < 0.05; ** *p* < 0.01; *** *p* < 0.001.

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
