# Peer review of "Characterisation and In Vitro Evaluation of Fenugreek (Trigonella foenum-graecum) Seed Gum as a Potential Prebiotic in Growing Rabbit Nutrition"

_animals, 2020, doi:10.3390/ani10061041_

Round 1
Reviewer 1 Report
The suggestions for the authors are in the file that is attached

Author Response
Dear reviewer 1,
We have rewritten the document following all the recommendations of the reviewers. We are very grateful for the work of the reviewers as the document has improved with their evaluation.
Abstract
RV1: Abstract 40 Put "digestion in vitro
Author: Done
Introduction
RV1: Introduction L65-69 It makes references to other species, birds and pigs. The digestive physiology of birds is very different from that of rabbits and in our opinion the reference is dispensable. It is acceptable to refer to pigs, in the absence of studies on rabbits or rats.
Author: It is true that rabbit physiology and nutrition is different from those of birds and pigs, and perhaps it could be closer to pigs than birds. However, this sentence is aiming to highlight the potential prebiotic of galactomannan and why we are interested in it, for this reason we recommend maintaining this sentence.
RV1: 94-100 For some components the analysis reference is placed. Refer to the methodology used for these analyses or mention that it will be placed in the chemical analysis section.
Author: We agree with the reviewer. The following sentences was included: “Chemical composition methodology used is described at 2.3 section.”
Materials and methods
RV1: L130-133 the addition of soluble or insoluble fibre was done by addition to a basal diet, which led to the dilution of other ingredients. In the case of calcium carbonate and Premix should not they also be reduced? For ingredients with low content we suggest placing their inclusion with a decimal place.
Author: No, minerals and premix supply were maintained constant. It was described in the main text in the new version for a better understanding.
RV1: 143-177 As references are made to the methodologies used, we do not find it necessary to describe what has been done exhaustively. This part should be summarized and refer only to the most relevant and some adaptations if they have been effectuated.
Author: We agree with the reviewer, but in opinion, this section was written citing the origin of the methodology and only describing that not included in the cites and relevant for an adequate repeatability of the methods by the readers.
RV1: 179-181 And 197-215, We question whether there is a need to put a different reference for each component. In the effort to reduce the references of the work, can some of them be used for more than one analysis?
Author: In this part we made sure that every methodology be mentioned with its references, it will be easier for us to use one references for more than one methodology unfortunately its not the case. But I believe that in other sections we could reduce the number of references.
RV1: 220-222, This sentence should be placed elsewhere in the methodology.
Author: in the L 220-222 we describe calculations of our measured parameters used during the statistical analysis. We prefer its inclusion in this section as gather calculations from different methodological sections (digestion, fermentation…), but we are open to move this section to the previous one.
Results and discussion
RV1: L241: The data in table 3 refer to another subject. You should make a paragraph.
Author: We agree with the reviewer, it was changed to the beginning of the following paragraph where results of this table 3 are described.
Rv1: L244 the values are in the table. To facilitate hand reading they should be repeated
Author: We agree with the reviewer, we have deleted this sentence to avoid repetitions.
RV1: L246 Idem to line 241
Author: We agree with the reviewer, it was changed to the beginning of the following paragraph where results of this table 4 are described
RV1: L280 The data for unfermented fraction do not coincide with those in the table, where a decrease of 83 is observed.
Authors: In the table we put the value of the unfermented fraction of the contrast SF-IF (-83.1), but in the text we talk about the fermentation of the indigestible fraction for a better understanding, and it is the opposite (+83 g/kg).
Conclusion
RV1: 391-402 An effort should be made to summarise this part, leave only the essentials and not repeat the results and discussion
Author: Maybe we can change this part from this:
“In the present work, we have tried to evaluate the potential of an extract of FSG, rich in galactomannan, as a functional product with prebiotic potential for growing rabbits. In this way, FSG satisfies two essential conditions of the prebiotic effect; it shows resistance to in vitro enzymatic digestion and was totally in vitro fermented by caecal bacteria. Consequently, pure FSG promoted the caecal fermentation as expected for a prebiotic, reducing caecal pH and N-NH3 content, increasing total VFA production and modulating VFA profile. However, when the FSG was included in prebiotic doses in rabbit feeds the results were not entirely satisfactory. FSG linear inclusion up to 20 g/Kg in growing feeds led to a reduction in the solubility of some nutrients during in vitro digestion, especially in enriched soluble fibre diets, and although FSG increased the fermentation fraction of the diets, no relevant changes in the fermentation profile were observed. For this reason, it is necessary to perform in vivo studies with different sources of fibre to truly determine the prebiotic potential of FSG rich in galactomannan.”
To this:
“In the present work, we have tried to evaluate the potential of an extract of FSG, rich in galactomannan, as a functional product with prebiotic potential for growing rabbits. In this way, FSG satisfies two essential conditions of the prebiotic effect; it shows resistance to in vitro enzymatic digestion and was totally in vitro fermented by caecal bacteria. Consequently, pure FSG promoted the caecal fermentation as expected for a prebiotic, reducing caecal pH and N-NH3 content, increasing total VFA production and modulating VFA profile. However, FSG linear inclusion up to 20 g/Kg in growing feeds led to a minor no relevant effects on nutrients solubility and the fermentation profile. For this reason, it is necessary to perform in vivo studies with different sources of fibre to truly determine the prebiotic potential of FSG rich in galactomannan.”
Authors: We have modified the conclusion following the recommendations of the reviewer.
Tables
RV1: Tables 3 and 5 In the average values, we suggest to standardize the number of decimal places and adjust according to the order of magnitude. In the order of tens and hundreds, one decimal place, in the order of units, two decimal places and infer to 0, 3 decimal places. The value of SEM must one more decimal place than the average values.
Author: We agree with the reviewer. In Table 3, we have maintained the current presentation with 4 relevant figures, but we have tried homogenising the number of decimals in Table 5, with the same decimals per row, but maintaining the decimals required to explore the significant contrasts presented.
References
RV1: The paper presents 64 bibliographic references. It should be made an effort and reduce to 40 to 45. Preference leave quotes from international journals and conferences related wit this work. For some statements using 5 or 3 references (line 59 and 62), you should leave only 1 or 2, the ones that suit to work.
Author: We agree with the reviewer, we must try to use only the number of references required. However, in the case of products characterization and evaluation, due to the large number of different analyses that the product and diets are subject, it is complicated to reduce the number of references for a good methodological repetition and maintain an adequate justification and discussion of the work. In fact, main of our arguments has been supported by one reference. For this reason, we recommend maintaining the current list of references.
RV1: 420 References should be standardised. With the same number of authors (2 or 3), in some situations "and" is placed and in others not. For example, see references 4, 8, 12, 14 or 28 and 29. RV1: 424 In some references an "&" is placed in the link to the last author (5) and in others not. Author: the “&” and the “and” before the last author in al references are eliminated in order to have the same structure in L414, L419, L420, L424, L431, L434, L437, L453, L490.
Author: We agree with the reviewer, refrences are standardised according to the current journal recommendation.

Reviewer 2 Report
The paper describes a study aiming at evaluating the characteristics of an extract of fenugreek rich in galattomannan and its in vitro evaluation as a prebiotic for rabbit feeding.
The introduction is pertinent and updated. The experimental design is correct and the methodologies applied for extract characterization and in vitro evaluation are clearly described; the number of samples for chemical analyses is adequate as well as the number of replicates for in vitro analyses.
The experimental diets covered the feeding standards for rabbits in the post-weaning phase. The rabbits used for caecal content sampling were treated in the respect of animal welfare. The results are clearly presented and correctly discussed. The conclusions and implications are supported by the results.
Apparently the text presents some grammatical and syntax errors (but I am not an English mother tongue) and should be revised.
The paper is worth to be published after a moderate revision according the suggestions, mainly formal, below.
- Line 2 (title): put “in vitro” in italic all over the manuscript.
- L17 and whatever in the manuscript: change “Kg” to “kg”.
- L28: delete “also”.
- L30, L232 and L319: a content of CP of 223 g/kg can be considered “rather high” or “moderate”, but not “non-negligible”.
- L31 and whatever in the manuscript: the enzymatic phase is called “digestion” or “pre-digestion…” and in other ways and this can confuse the reader, together with the terms “digestible fraction” and “undigestible fraction”. The authors should be more precise when they mention the “enzymatic phase” and the “fermentation phase” of “in vitro digestion” that is the sum of the two phases. The same with the terms “soluble” and “solubility”, that are used sometimes in a not appropriate way.
- L31: the proportion of digested FSG should be expressed as % instead of 145 g/kg that may be confused with the concentration of substances in the extract. Modify the sentence as: “…affected by in vitro digestion, as only 14.5% of FSG was dissolved during the enzymatic phase”.
- L32: modify as: “The linear inclusion of FSG up to 20 g/kg”
- L33-34: the sentence should be clarified as: “The fraction not digested during the enzymatic phase of pure FSG almost completely disappeared (98.4%) during the subsequent fermentation phase.
- L39: delete “really”.
- L48: maybe “increases” should be correct as “increase”.
- L49-51: the final part of the sentence is not clear “a minimum level …. being required…” and not well related to the previous part. Please rewrite.
- L52-53: “Some of the substances that constitute the soluble fibre affect …”
- L73: put in italic “Trigonella foenum-graecum”.
- L84: “…were subjected to “.
- L95-98: references for these analyses should be provided or the text should refer to a following section (see 2.3 Chemical analyses).
- L107: “the rabbit caecal inoculum”.
- L155: change “11 fraction” to “11 substrates”.
- L157-161: rearrange the sentences “.. until slaughter. The rabbits had ad libitum access……antibiotic treatment. At 56 d of age, they were sacrificed …..aerobic environment. Under constant CO2….
- L187: insert space after the point.
- L194-195: “[41] by AOAC [39] …….. Van Soest [42], with ….”
- L233 (footnote table 2): “but gel production during analysis made its determination uncertain”.
- L252-253 and L345: see previous comment on L31.
- Figure 1: the colors of the 3 parts of the bars are not clearly identified in the title. The letters used for statistical differences do not help to identify the three portions (enzymatically digested, fermented, neither digested nor fermented).
- Figure 2: a spiral sign before “?neutral detergent fibre” does not appears in the figure (it should be a triangle).
- Table 5 (last row): the negative value (-28.89) for N-NH3 is likely wrong.
- L330 and L336: likely the two citations of “Trocino” e al.” have to be swapped; [54] on L336 and [6] on L330.
- L354-367. Please homogenize the form of “ic” or “ate” for VFA names (propionic acid or propionate…). Do not use propionic, valeric, …. as nouns. All over the manuscript.
- L367: “The higher proportions of isobutyric and isovalerc acids ….
- L392: change “prebiotic potential” by “prebiotic activity”
- L401: with in vivo trial you should determine the “prebiotic effect or efficacy”, not only the “potential”.
- Reference list: the references should be carefully checked because they are formatted in different styles and forms, not always corresponding to the author’s rules. There are errors in spaces, commas and semicolon, abbreviations or not of the journal name. The last name of the authors is preceded by “and” (e.g. L414), “&” (e.g. L424), semicolon (ref. 419) or nothing.
- L414: insert a space before “and”. Insert a point before “Cuniculture”.
- L419: insert the title of the paper, year number and pages.
- L423: change commas between names with semicolons.
- L443-L446, L512 and other references: insert spaces before the titles.
- L490: Correct De Blas (not De Bias). Insert final page. You can also cite the 3rd edition of the CABI book (2020).
- L440: the date of the congress is lacking (also in other references).
- L532: check the capital letter for authors and editor’s names.
- L539: check colon and semicolon between names; Javier Garcia should be Garcia J.
Author Response
Dear reviewer 2,
We have rewritten the document following all the recommendations of the reviewers. We are very grateful for the work of the reviewers as the document has improved with their evaluation.
RV2: Line 2 (title): put “in vitro” in italic all over the manuscript.
Author: L2 all over the text “in vitro” is changed in italic
RV2: L17 and whatever in the manuscript: change “Kg” to “kg”.
Author: L17 and in all the text “Kg” is replaced by “kg”
RV2: L28: delete “also”.
Author: “also” is deleted in L28
RV2: L30, L232 and L319: a content of CP of 223 g/kg can be considered “rather high” or “moderate”, but not “non-negligible”.
Author: L30, L232 and L319, changes are made from “non-negligible content protein to moderate protein content.
RV2: L31 and whatever in the manuscript: the enzymatic phase is called “digestion” or “pre-digestion…” and in other ways and this can confuse the reader, together with the terms “digestible fraction” and “undigestible fraction”. The authors should be more precise when they mention the “enzymatic phase” and the “fermentation phase” of “in vitro digestion” that is the sum of the two phases. The same with the terms “soluble” and “solubility”, that are used sometimes in a not appropriate way.
Author: We agree with the authors and it has been changed in the whole text by enzymatic digestion or enzymatic phase.
RV2: L31: the proportion of digested FSG should be expressed as % instead of 145 g/kg that may be confused with the concentration of substances in the extract. Modify the sentence as: “…affected by in vitro digestion, as only 14.5% of FSG was dissolved during the enzymatic phase”. RV2: L252-253 and L345: see previous comment on L31.
Author: In animal nutrition is widely recommended the use of g/kg respect to %, for this reason we prefer the current presentation of these values.
RV2: L32: modify as: “The linear inclusion of FSG up to 20 g/kg”
Author: the change requested by the reviewer has been done in L32.
RV2: L33-34: the sentence should be clarified as: “The fraction not digested during the enzymatic phase of pure FSG almost completely disappeared (98.4%) during the subsequent fermentation phase.
Author: We have modified the sentence following the recommendations of the reviewer.
RV2: L39: delete “really”.
Author: “really” is deleted from L39
RV2: L48: maybe “increases” should be correct as “increase”.
Author: Done
RV2: L49-51: the final part of the sentence is not clear “a minimum level …. being required…” and not well related to the previous part. Please rewrite.
Author: the sentence has been rewritten as “It is well known that a deficit in fibre or an excess of protein increase the risk of digestive disorders [4-5], and that the level and nature of the fibre can also affect the digestive health of weaned rabbits, thus a minimum level of both insoluble and soluble fractions of fibre are required to reduce the digestive risk [3].”
RV2: L52-53: “Some of the substances that constitute the soluble fibre affect …”
Author: modification is done as the reviewer requested.
RV2: L73: put in italic “Trigonella foenum-graecum”.
Author: in L73 “Trigonella foenum-graecum”. In modified in italic.
RV2: L84: “…were subjected to “.
Author: modification is done from “were subject to” to “were subjected to”.
RV2: L95-98: references for these analyses should be provided or the text should refer to a following section (see 2.3 Chemical analyses).
Author: Done
RV2: L107: “the rabbit caecal inoculum”.
Author: “the caecal inoculum” is modified to “the rabbit caecal inoculum”
RV2: L155: change “11 fraction” to “11 substrates”.
Author: ok, “11 fraction” is modified to “11 substrates”
RV2:L157-161: rearrange the sentences “.. until slaughter. The rabbits had ad libitum access……antibiotic treatment. At 56 d of age, they were sacrificed …..aerobic environment. Under constant CO2….
Author: L157-161: Done.
RV2: L187: insert space after the point.
Author: the space after the point was inserted
RV2: L194-195: “[41] by AOAC [39] …….. Van Soest [42], with ….”
Author: modification are done in L194-195 as requested
RV2: L233 (footnote table 2): “but gel production during analysis made its determination uncertain”. Author: the phrase is modified as requested
RV2: Figure 1: the colors of the 3 parts of the bars are not clearly identified in the title. The letters used for statistical differences do not help to identify the three portions (enzymatically digested, fermented, neither digested nor fermented).
Author: Done, we have described better the sections and colours of the figure and the letters for statistical differences.
RV2: Figure 2: a spiral sign before “?neutral detergent fibre” does not appears in the figure (it should be a triangle).
Author: Solved
RV2: Table 5 (last row): the negative value (-28.89) for N-NH3 is likely wrong.
Author: Corrected
RV2: L330 and L336: likely the two citations of “Trocino” e al.” have to be swapped; [54] on L336 and [6] on L330.
Author: the references were switched as requested
RV2: L354-367. Please homogenize the form of “ic” or “ate” for VFA names (propionic acid or propionate…). Do not use propionic, valeric, …. as nouns. All over the manuscript.
Author: Homogenised
RV2: L367: “The higher proportions of isobutyric and isovaleric acids ….
Author: changes to iso-butyrate and iso-valerate.
RV2: L392: change “prebiotic potential” by “prebiotic activity”
Author: L392 “prebiotic potential” was modified to “prebiotic activity”
RV2: L401: with in vivo trial you should determine the “prebiotic effect or efficacy”, not only the “potential”.
Author: The “potential” was replaced by the word “efficiency”
References
RV2: Reference list: the references should be carefully checked because they are formatted in different styles and forms, not always corresponding to the author’s rules. There are errors in spaces, commas and semicolon, abbreviations or not of the journal name. The last name of the authors is preceded by “and” (e.g. L414), “&” (e.g. L424), semicolon (ref. 419) or nothing.
Author: references are homogenised, without “and” or “&”, and authors are separated with semicolons.
RV2. L414: insert a space before “and”. Insert a point before “Cuniculture”.
Author: Ok done for the point before “Cuniculture”, the “and” is deleted. In all references “and” was deleted and replaced by a semicolon. Commas between authors were replaced by semicolons. All spaces between authors first names abbreviation were eliminated. All dates after the list of authors were eliminated two.
RV2: L419: insert the title of the paper, year number and pages.
Author: Solved.
RV2: L423: change commas between names with semicolons.
Author: All commas between names were modified to semicolons
RV2. L443-L446, L512 and other references: insert spaces before the titles.
Author: Spaces were inserted before the title in al references
RV2: L490: Correct De Blas (not De Bias). Insert final page. You can also cite the 3rd edition of the CABI book (2020).
Author: L490 De Blas was corrected; the final page was added.
RV2: L440: the date of the congress is lacking (also in other references).
Author: L436, L438, L440 the date of the conference was added
RV2: L532: check the capital letter for authors and editor’s names.
Author: L532 modification was done as requested
RV2: L539: check colon and semicolon between names; Javier Garcia should be Garcia J.
Author: L539 modification was done as requested
